# Identification of Autoantibodies to a Hybrid Insulin Peptide in Type 1 Diabetes

**DOI:** 10.3390/diagnostics13172859

**Published:** 2023-09-04

**Authors:** Janet M. Wenzlau, Yong Gu, Aaron Michels, Marian Rewers, Kathryn Haskins, Liping Yu

**Affiliations:** 1Department of Immunology and Microbiology, University of Colorado School of Medicine, 12800 East 19th Avenue, Mail Stop 8333, Aurora, CO 80045, USA; janet.wenzlau@cuanschutz.edu (J.M.W.); katie.haskins@cuanschutz.edu (K.H.); 2Barbara Davis Center for Childhood Diabetes, 1775 Aurora Court, Mail Stop B140, Aurora, CO 80045, USA; yong.gu@njmu.edu.cn (Y.G.); aaron.michels@cuanschutz.edu (A.M.); marian.rewers@cuanschutz.edu (M.R.)

**Keywords:** type 1 diabetes, autoantibodies, post-translational modification, β-cell, secretory granule protein, biomarker

## Abstract

Type 1 diabetes (T1D) is a chronic autoimmune disease that attacks the insulin-producing b cells of the pancreatic islets. Autoantibodies to b cell proteins typically appear in the circulation years before disease onset, and serve as the most accurate biomarkers of T1D risk. Our laboratory has recently discovered novel b cell proteins comprising hybrid proinsulin:islet amyloid polypeptide peptides (IAPP). T cells from a diabetic mouse model and T1D patients are activated by these hybrid peptides. In this study, we asked whether these hybrid molecules could serve as antigens for autoantibodies in T1D and prediabetic patients. We analyzed sera from T1D patients, prediabetics and healthy age-matched donors. Using a highly sensitive electrochemiluminescence assay, sera were screened for binding to recombinant proinsulin:IAPP probes or truncated derivatives. Our results show that sera from T1D patients contain antibodies that bind larger hybrid proinsulin:IAPP probes, but not proinsulin or insulin, at significantly increased frequencies compared to normal donors. Examination of sera from prediabetic patients confirms titers of antibodies to these hybrid probes in more than 80% of individuals, often before seroconversion. These results suggest that hybrid insulin peptides are common autoantigens in T1D and prediabetic patients, and that antibodies to these peptides may serve as valuable early biomarkers of the disease.

## 1. Introduction

Autoantibodies against pancreatic islet β-cell antigens have been well established as the most reliable biomarkers for risk of type 1 diabetes (T1D) [1]. The appearance of circulating islet autoantibodies (IAbs) targeting any of four major molecularly defined antigens (insulin, islet antigen 2 (IA-2), glutamic acid decarboxylase 65 (GAD-65) and zinc transporter 8 (ZnT8) [2,3,4,5,6]) typically precedes overt disease by years. Insulin autoantibodies (IAAs) are frequently the first to emerge in young children, followed sequentially by any of the other IAbs, depending on age. At clinical onset, 95% of patients have IAbs to at least one of these β-cell proteins, and the cumulative measure of antigen specificities functions as a prospective risk indicator for progression to T1D [1,6,7]. The combined measure of the common IAbs is not sufficient to identify all cases of T1D, as a small proportion of patients (2–5%) present as IAb-negative at diagnosis, suggesting that other antigens have yet to be defined [2]. 

An emerging source of potentially valuable biomarkers in T1D stems from the immunological impact of exogenous viral and bacterial antigens, and post-translational modifications (PTMs) of native proteins that may escape immune tolerance [8,9,10]. *Mycobacterium avium* subspecies *paratuberculosis* (MAP) is associated with a variety of autoimmune diseases, including T1D, possibly via molecular mimicry of MAP peptides that cross-react to sequences in proinsulin, GAD-65, and ZnT8 [9,10]. Enteroviruses have long been proposed to induce T1D by various mechanisms (including molecular mimicry) [11], and human endogenous retroviral (HERV) proteins have been shown to act as antigens for humoral immunity. A correlation between the presence of anti-HERV-W envelope protein antibodies and IAAs has been reported among T1D individuals. Additionally, a higher prevalence of anti-HERV-K envelope antibodies has been reported in the sera of T1D patients compared to healthy controls [10]. A number of PTMs have been reported to induce T cell immunoreactivity in T1D, including oxidation, deamidation, transglutamination, citrullination, and aberrant secondary structure [8,12,13], all of which may increase binding affinity to disease-associated HLA binding grooves. The predominant T1D antigens for B cells are also established T cell antigens and, in some cases, the antigenic epitopes overlap [14,15]. PTMs of established antigens, such as GAD65 and insulin, have been shown to yield neoantigenic epitopes for IAbs that have predictive value for disease risk [16,17,18].

A newly reported PTM, unique in that it is independent of chemical or enzymatic conversion of amino acids, involves junctional association of proinsulin fragments with proteolytic cleavage products of other granule proteins to form hybrid insulin peptides (HIPs). These hybrid peptides have been identified in mouse and human pancreatic islets [19]. Human T cells respond to a range of HIPs containing C-peptide truncations fused to the natural cleavage products of islet secretory granule proteins, including islet amyloid polyprotein (IAPP), neuropeptide Y 1 (NPY1), or insulin A chain [20,21], among others [19]. In each reported case, the C-peptide component of the HIP is truncated at either amino acid 47 or 58 (relative to proinsulin).

It is unknown whether circulating anti-HIPs IAbs exist in T1D. To address this possibility, we generated HIP antigens based on T cell epitope studies, and monitored patient sera for anti-HIPs IAbs. For this work, we exploited a modified exceptionally sensitive electrochemiluminescent (ECL) antibody assay that has been used by others for early detection of insulin antibodies [22]. The results show that IAbs against complex 3D HIP antigens exist in T1D patients, and arise early in at least 80% high risk pre-T1D patients, with their detection frequently preceding that of insulin antibodies. These results suggest that anti-HIPs IAbs are relevant components of T1D, and potential early biomarkers of this disease.

## 2. Materials and Methods

### 2.1. Patient Samples and Human Ethics

#### 2.1.1. New-Onset T1D Participants

Sera collected from T1D patients within two weeks of diagnosis at the Barbara Davis Center were selected on the basis of expressing IAbs targeting at least one of the following proteins: insulin, GAD65, IA-2, or ZnT8. Among the new-onset T1D individuals, 48% (129/269) were also IAA^+^ and 79% (212/269) were multiple IAb^+^.

#### 2.1.2. Non-HLA-Matched Controls

The Autoimmunity Screening for Kids (ASK) program is a general population screening for preclinical T1D (and celiac disease) using the IAb biomarkers targeting insulin, GAD65, IA2, ZnT8, and a celiac disease autoantibody biomarker recognizing transglutaminase. The goal of the study is to identify children (aged 2–17 years) at risk of developing the above autoimmune diseases, and to reduce the incidence of diabetic ketoacidosis.

#### 2.1.3. Prediabetic Individuals

The Diabetes Autoimmunity Study in the Young (DAISY) prospectively follows children with predisposing genetic risk (HLA-DR/DQ genotypes) for development of IAbs, and children who are first-degree relatives (FDRs) of T1D individuals. Informed consent was obtained from each individual (of legal age) or parent of each subject, and all research protocols were approved by the Colorado Multiple Institutional Review Board in accordance with National Institute of Health (USA) guidelines.

### 2.2. Recombinant Hybrid Probes

Figure 1 shows the full-length proinsulin and pro-IAPP sequences, including the IAPP insertion sites within proinsulin (left) and cleavage site for the IAPP peptide (right) (Figure 1A). Schematics of all probe constructs are shown in detail in Figure 1B.

Linear HIP peptides (Figure 1B Top) were synthesized at >80% purity (Peptide 2.0, Chantilly, VA, USA) and resuspended at 2 mg/mL in 2XPBS. Cyclic C47/IAPP HIP peptides were synthesized at >98% purity (Peptide 2.0, Chantilly, VA, USA), and peptides were reconstituted at 2 mg/mL in PBS.

Production of recombinant three-dimensional HIPs (Figure 1B) was outsourced (Creative BioMart, Shirley, NY, USA). Double-stranded DNAs encoding three-dimensional (3D) HIPs were generated using complementary overlapping PCR primers (Integrated DNA Technologies, Coralville, IA, USA), with unique restriction enzyme sites at the 5′ and 3′ ends for directional cloning into pcDNA3.1.

DNA was subcloned into vectors for bacterial expression and purification (>90% purity). PI HIP recombinant proteins were denatured and refolded according to a proprietary protocol, and resuspended in a buffer (20 mM Tris-HCL (pH 8.0), 150 mM NaCl) at 0.5 mg/mL. Recombinant human insulin (Humulin U-100, 100 units/10 mL) was purchased from Eli Lilly, (Indianapolis, IN, USA). HPLC-purified (>99%) human proinsulin was purchased from AmideBio (Louisville, CO, USA).

### 2.3. Labeling of Recombinant Hybrid Probes

Labeling of recombinant probes was performed by conjugation of Meso Scale Discovery (MSD) Gold sulfo-TAG NHS-Ester and biotin to primary amines, as previously described [23]. Briefly, recombinant HIP protein was mixed with sulfo-TAG, incubated at RT for 2 h, and purified on a desalting spin column (Thermo Scientific, Waltham, MA, USA). Labeling efficiency was determined by spectrophotometry (450 nm), and protein concentration by a bicinchoninic acid (BCA) kit (Sigma Aldrich, St. Louis, MO, USA). Biotin labeling of PI HIP recombinant protein was performed using a biotinylation kit (Thermo Scientific), and products purified with a desalting spin column. Protein concentration and labeling efficiency were determined by spectrophotometry (500 nm) and a BCA kit, respectively.

### 2.4. Electrochemiluminescence (ECL) Assay

The ECL antibody assay has been reported previously [20]. For the HIP ECL IAb assay, parameters were systematically optimized for concentrations and ratios of biotinylated and sufo-TAG recombinant HIP antigen probes, serum volume, and serum concentration. Briefly, freshly labeled sulfo-TAG HIP recombinant protein (300 ng/mL) and biotin-labeled recombinant HIP protein (400 ng/mL) were combined with serum in PBS (5% BSA). Mixtures were incubated for 2 h at RT with shaking, and then overnight at 4 °C. Streptavidin-coated ECL plates were blocked with 3% MSD blocker “A” O/N at 4 °C. ECL plates were washed 3× with PBST before addition of the antigen probe/serum mixtures, which were incubated for 1 h at RT and washed 3× with PBST. HIP IAbs were quantified by counting on a SQ120 MSD instrument. A highly positive and negative serum were used as the internal positive and negative controls, respectively. The cut-off value of 300 counts per second (CPS) was set for the assay, according to the highest background not absorbable by HIP protein in the preabsorption assay. Each serum sample was assayed in duplicate and averaged. Assays were performed three times per sera sample.

### 2.5. IAb Preabsorption (Blocking) ECL Assay 

To determine the specificity of binding in the HIP-ECL assay, sera samples (*n* = 21) with different anti-HIP IAb titers from distinct cohorts (10 new onset T1D, 7 DAISY post-onset, 2 healthy control, 2 T2D) were randomly selected for preabsorption testing. Serum diluted 5-fold in PBS (20 μL) was mixed with one of the following unlabeled proteins: cyclic PI C47/IAPP, cyclic C47/IAPP peptide, or C47/IAPPHIP recombinant protein at a concentration of 2 μg/mL, as previously described [24]. Freshly prepared labeled (biotin and sulfo-TAG) C47/HIP recombinant protein (20 μL) in PBS containing 5% BSA was then added, and the assay performed as described above. PBS containing 5% BSA was used as a negative control. Each serum was preabsorbed in duplicate within the assay, and measured in three experiments.

### 2.6. IAb Affinity Assay

Serum samples containing PI HIP IAb were diluted 5× with PBS (20 μL/ea), and incubated with increasing concentrations (0.25 μg, 0.5 μg, 1 μg, 2 μg) of unlabeled recombinant protein C47/IAPP and PI C58/IAPP. Then, 20 μL of freshly prepared labeled PI HIP recombinant antigen probe in PBS containing 5% BSA was added, and the HIP-ECL assay was performed as described above. PBS alone was used as a negative control. Each serum was assayed in duplicate and measured in three experiments.

### 2.7. Statistics

The incidence and levels of IAbs and HIP-specific antibodies were compared between patients and healthy controls. Statistical analyses were performed using Fisher’s exact test for comparison of the incidences, and rank-sum test for comparison of autoantibody levels using PRISM 4·0 version software (GraphPad Software Inc., San Diego, CA, USA). A two-tailed *p*-value with an alpha level for significance was set at 0.05. Welch’s *t*-test was used in cases of unequal variances (Figure 2B). Although related to the standard Student’s *t*-test, this test is used to test the hypothesis that two populations have equal means. The results are very similar to the Student’s *t*-test unless the group sizes and standard deviations are very different.

## 3. Results

### 3.1. HIP Probes Are Recognized by IAbs in T1D Patients

#### 3.1.1. HIP Epitopes Are Conformational

In initial studies, we developed linear HIP antigen probes that mimic human HIP T cell epitopes, as previously described [19,20]. Studies have shown that HIPs are derived from the cleavage product IAPP, which is located adjacent to the insulin C-peptide moiety at either amino acid 47 or 58 (relative to proinsulin) [20]. We generated a dual HIP probe containing IAPP adjacent to C-peptide at either site (see Figure 1), separated by a linker. The probe was designed as a dimer to potentially amplify IAb binding while simultaneously limiting background binding.

In direct binding ECL assays, sera from 95 T1D patients positive for at least one “gold standard” IAb and age-matched control sera from a healthy cohort of the general population (Autoimmunity Screening for Kids (ASK) study, *n* = 96) were tested for linear peptide recognition (see Table 1, cohorts). Comparisons revealed no significant increased peptide binding by sera from T1D patients (Figure 2A).

#### 3.1.2. Complex HIP Probes Bind IAbs in T1D Patients

The mechanism(s) of HIP antigen and intermediate molecule formation have not been defined. Mass spectrometry analysis of proteins from mouse and human islet preparations have been performed with proteolytically cleaved peptides to improve spectra with shorter peptides [20]. Undigested mouse islet protein preparations analyzed by mass spectrometry are typically smaller than 30 amino acids, including the C58/IAPP HIP [25], which may be a fragment of the endogenous protein intermediate. Studies to define the mechanism of HIP formation suggest the involvement of cathepsin D and cathepsin L in the b cell granules and other organelles, although these analyses were performed in vitro with truncated peptide moieties [26,27]. The structure of the native intermediate HIPs is unresolved.

As many antibody epitopes are conformational and require a 3D structure, we generated more complex proinsulin-based HIP antigens and tested their binding to potential IAbs within T1D patient sera. The “closed loop” HIP probe was constructed by inserting IAPP at either proinsulin (PI) amino acid (aa) position 47 or 58 within the C-peptide sequence, and the subsequent recombinant fusion proteins were folded to achieve the correct secondary structure (Figure 1, see Materials and Methods). Recombinant HIP proteins were separately tagged with either biotin and sulfo-TAG, and used in an indirect ECL binding assay to capture serum IAbs (see experimental design, Figure 2B).

Sera from newly diagnosed T1D patients (*n* = 269), T2D patients (*n* = 112), and healthy controls (ASKs, *n* = 96) were compared for binding to both c47/IAPP and c58/IAPP probes. Scatter plots revealed that sera from 40% T1D and 26% T2D patients had significant levels of IAbs targeting the recombinant HIP antigen probes (Figure 2C, Welch’s *t*-test for unequal variances showed the same statistical significance when comparing T1D and control values). Comparison of sera from high- or low-risk HLA genotyped T1D patients showed no significant differences (Figure 2D). Within the high-risk cohort, 42% (37/89) and 83% (74/89) of individuals were IAA^+^ and multiple IAb^+^, respectively. In the low-risk cohort, 46% (39/84) were IAA^+^ and 75% (63/84) were multiple IAb^+^.

#### 3.1.3. Conformational HIPs Exclusively Block IAb Binding

To determine which HIP IAPP insertion site (C47 or C58) contributes to IAb specificity, sera from selected T1D patients were tested in an inhibition ECL assay (see Figure 3A for details). In this case, sera were preincubated with unlabeled C47/IAPP (solid lines) or C58/IAPP (dotted lines), followed by immunoprecipitation with labeled probe as defined (Figure 3B). The results showed that preabsorption with C47/IAPP resulted in significantly reduced immunoprecipitation of both probes by all sera, whereas preabsorption with C58/IAPP resulted in partial or no blocking (Figure 3B). These results indicate that the HIP C47 insertion site was typically more important for IAb recognition.

To further elucidate IAb epitopes, smaller cyclic peptides lacking the complete proinsulin A and B chains that form the native disulfide bonds were generated and tested for inhibition of binding to the parent C47/IAPP antigen (see Figure 1B for constructs). In all cases, preabsorption with C47/IAPP (filled circles) but neither cyclic peptide blocked binding of the C47/IAPP probe to the HIP IAbs (Figure 3C). In parallel studies, preincubation with insulin, proinsulin, linear, or C47/random insert probes also failed to block C47/IAPP binding (Figure 3D and Appendix A). These combined results suggest that sera from T1D patients maintain circulating IAbs against complex HIP antigens. Furthermore, they imply that IAb binding requires an HIP structure with IAPP inserted at proinsulin aa 47 and full length proinsulin protein.

### 3.2. High Risk Pre-T1D Patients from DAISY Studies Have Early Titers of Anti-HIP IAbs

The predictive value of C47/IAPP HIP IAbs as biomarkers for T1D was evaluated using longitudinal sera samples from children at high risk of developing T1D, followed from birth to T1D onset (see Table 1). In the DAISY study, children who were first-degree relatives (FDRs) of T1D individuals or genetically predisposed to T1D were screened for seroconversion at approximately six-month to one-year intervals from birth and subsequently, for ‘gold standard’ autoantibody status, until clinical onset of T1D (ClinicalTrials.gov Identifier: NCT03205865, Figure 4A).

C47/IAPP IAbs were measured retrospectively using the direct ECL assay on sera from 12 individuals enrolled in the DAISY study who progressed to T1D. Studies revealed that all but two had HIP IAbs in their sera (83%), and nine patients had anti-HIP IAbs within the first 5 years (Figure 4B).

Longitudinal sera samples from 12 DAISY patients were then analyzed for the time of emergence of both HIP and insulin autoantibodies (Figure 4C–G). The results showed that sera from five patients contained HIP IAbs in the absence of (Figure 4C) or before insulin IAbs (Figure 4D), most commonly within the first year of study. In other patients, anti-HIP antibodies were detected simultaneously with or after anti-insulin IAbs (Figure 4E,F). Only 2 out of 12 patients appeared to have no significant anti-HIP IAb levels (Figure 4G). These studies confirm the presence of anti-HIP IAbs before disease onset and indicate that such IAbs may be valuable biomarkers for early prognosis.

## 4. Discussion

HIPs are part of a growing panel of immunogenic PTM antigens that are emerging as an important and logical explanation for the aetiology of T1D. Although T1D T cells have been identified that recognize specific HIP epitopes, it is unknown whether B cells and their secreted immunoglobulin receptors can also bind HIPs. This is an important concern because other T1D autoantibodies against components such as insulin are standard biomarkers for early detection of T1D. Here, we generated a series of antigens carrying HIP epitopes recognized by T1D T cells. Our results showed that sera from a significant number of T1D patients maintain circulating IAbs that recognize a complex 3D HIP structure containing the entire proinsulin sequence, including disulfide bonds. Most anti-HIP antibodies also preferentially bound HIPs containing the IAPP peptide inserted in proinsulin position aa 47 rather than aa 58, whereas either site can act as a T cell epitope. Thus, anti-HIP antibody epitopes appear to be more restricted than T epitopes, suggesting a specific 3D conformation is obligatory.

In this study, we observed a significant increase in the number of T1D patients harboring anti-HIP antibodies compared to age- and sex-matched controls, although 20% of controls also maintained anti-HIP antibodies at levels above background. It is a known phenomenon that many individuals maintain titers of low affinity autoantibodies, including those against known autoantigens such as DNA, although autoimmunity does not ensue [28]. Isolation and careful study of anti-HIPs antibodies from T1D patients and control subjects for relative binding affinities will be a future goal to address this important concern.

Individuals carrying the HLA_DQ2,8 haplotype are traditionally considered to be at highest risk for T1D [29], although recent work shows that 85% of new onset cases are spontaneous with no family history of the disease, and an ever-increasing proportion are not among the high-risk genotypes longitudinally monitored for the development of IAbs [30]. Our previous work showed that blocking with anti-DR, but not anti-DQ antibodies, completely abolished T cell responses against T1D-specific HIP peptides, indicating that HLA-DR is the most critical allele for anti-HIP T cell activation [19]. The fact that we did not observe any correlation between traditionally high (HLA_DQ2/8) and low (all others) risk HLA and anti-HIP IAbs in T1D patients in this study indicates that they require T help that is not DQ2/8-restricted. These results, and the conclusions of others [30], suggest a more complicated scenario that will require re-examination of restricting HLA determinants, particularly against nontraditional T1D peptides.

HIP IAbs are present in the sera of both T1D and T2D patients, and may be a common indicator of β-cell inflammation. Islet inflammation and immune cell infiltration are common and critical to both T1D and T2D [31]. T2D, which is closely correlated with increasing obesity, leads to insulin resistance and subsequent chronic islet inflammation to promote autoimmune responses. Clinical studies confirm the link between obesity and eventual onset of T1D [32]. Both IAbs and islet-specific T cells, hallmarks of T1D, have been identified in the circulation of some T2D individuals [33]. Overlapping pathologies for T1D and T2D imply common putative therapeutic interventions with efficacy for both diseases, such as anti-inflammatory remedies.

IAbs targeting post-translationally modified GAD, collagen, and insulin have all been detected in the serum of T1D patients, documenting the humoral immune response to PTM epitopes in T1D [16,18,32]. IAbs specific to oxidized insulin fulfil several of the criteria for new T1D biomarkers: they are a more sensitive biomarker than IAbs to native insulin, and they are present in most preclinical patients [18]. Our studies also predict IAbs against HIPs as relevant biomarkers. We have observed anti-HIP antibodies in most preclinical patients taken from DAISY studies and, importantly, detection of anti-HIP antibody precedes seroconversion by at least 1 year in more than 40% of DAISY patients. The characterization of human T1D anti-HIP antibodies, and their use to isolate in vivo circulating anti-HIPs, will be a future goal to further our understanding of the roles of HIPs in diabetes.

## Figures and Tables

**Figure 1 diagnostics-13-02859-f001:**
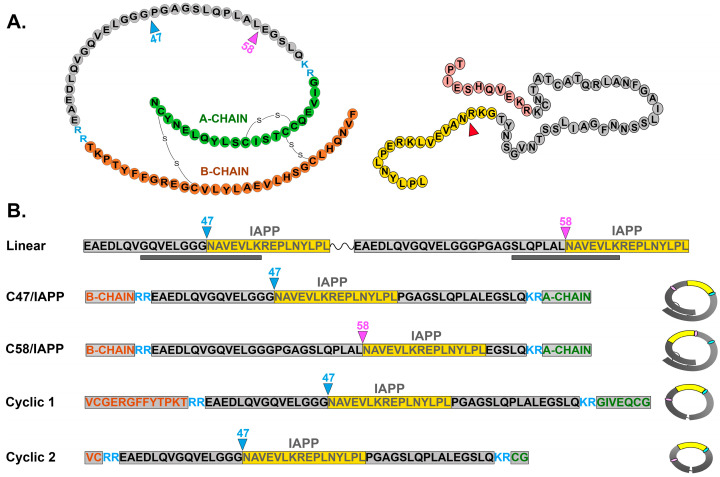
Schematic representation of proinsulin and proislet amyloid polypeptide (pro-IAPP) and recombinant probes used in this study. (**A**): Schematics of proinsulin (left), showing disulfide bonds and IAPP peptide insertion sites C47 and C58, and of pro-IAPP (right), showing cleavage site to generate C-terminal IAPP peptide for insertion into proinsulin C47 or C58. (**B**): Recombinant probes include (top) a linear probe containing two HIP peptides with IAPP (yellow) inserted at amino acid C47 or C58 of truncated proinsulin separated by a linker. Gray bars below indicate known T cell HIP epitopes. Conformational probes (below) include the whole proinsulin with IAPP inserted at either C47 or C58 (middle) or smaller probes (bottom), with or without partial A-chain and B-chain sequences (Cyclic 1 or Cyclic 2, respectively). Both cyclic peptides include terminal disulfide linkers. Figures at right show closed conformational structure for each probe.

**Figure 2 diagnostics-13-02859-f002:**
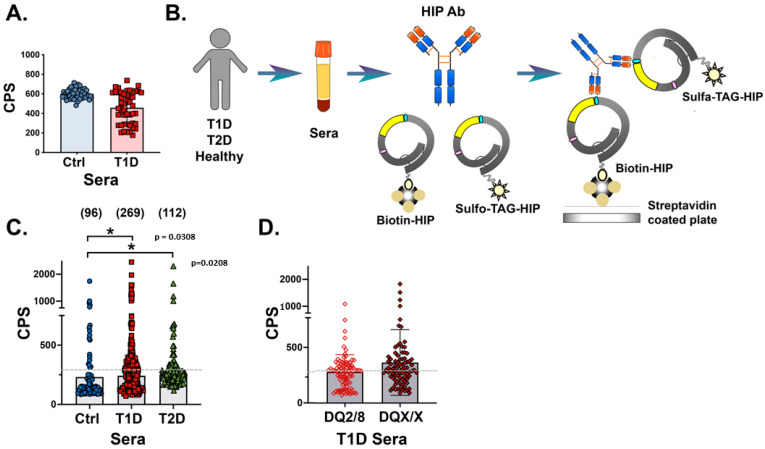
Investigation of HIP IAbs in T1D patient sera using linear and conformational antigen probes in a direct binding ECL assay. (**A**): Bar graph showing binding of sera from healthy first-degree relatives (FDRs) (blue circles, left) and T1D patients (red squares, right) to linear probe in radioimmunoassays. (**B**): Schematic of the direct binding ECL assay used to detect antibodies against the C47/IAPP probe. (**C**): Graph showing ECL immunoprecipitation of C47/IAPP probe using control (blue circles, left), T1D (red squares, middle), or T2D (green triangles, right) sera. Patient number in parentheses. (**D**): Direct ECL immunoprecipitation of C47/IAPP probe using T1D patient sera binned according to HLA high-risk (open diamond, left) or all other HLAs (solid diamond, right). Data are expressed as counts per second (CPS), and the mean indicated by shaded bar. Gray dashed lines indicate assay cut-off at 300 CPS. Each serum was measured in three assays in duplicate, and values averaged with representative data are shown.

**Figure 3 diagnostics-13-02859-f003:**
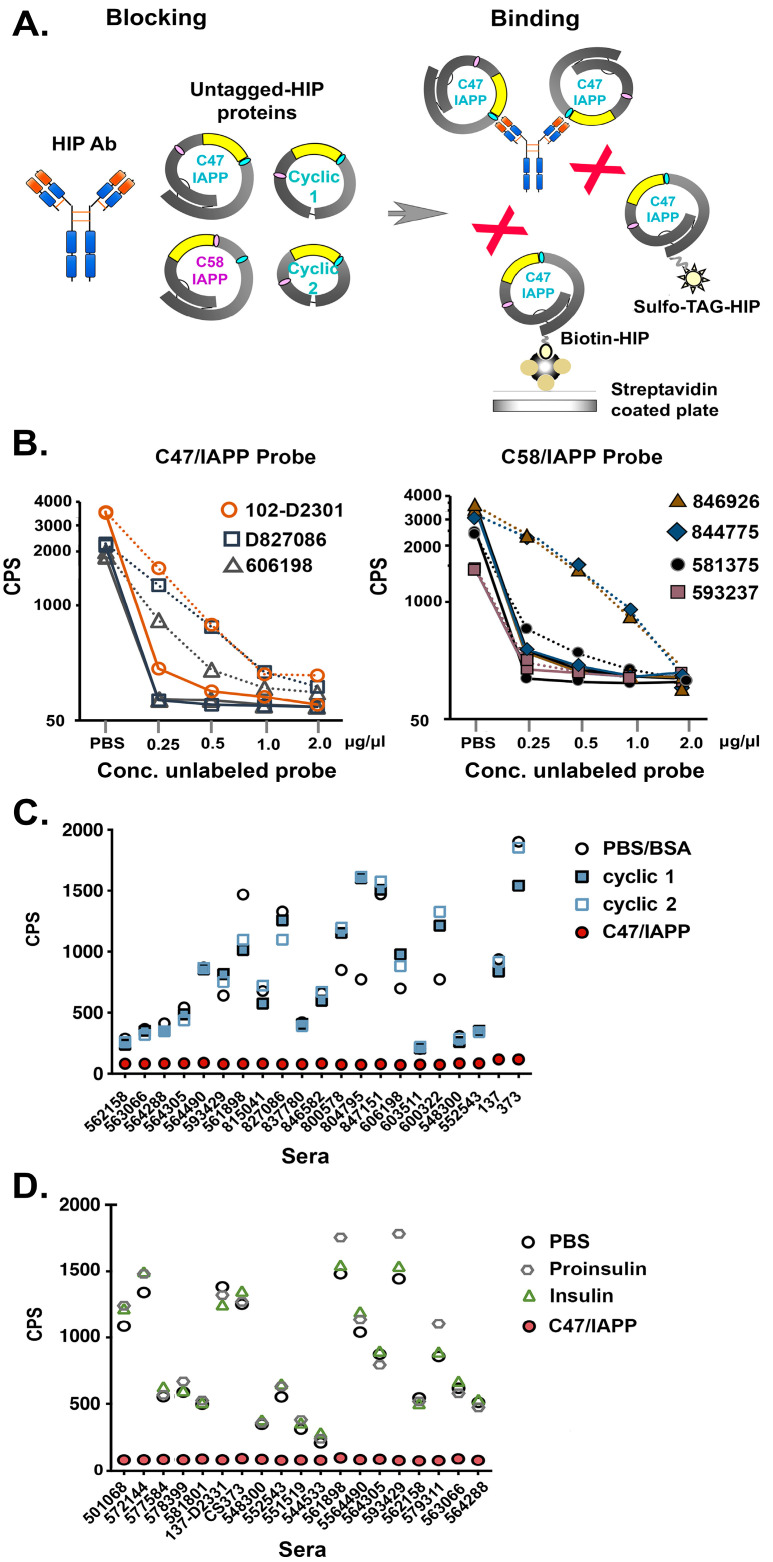
Investigation of HIP IAbs in T1D patient sera using defined antigen probes in a sensitive blocking ECL assay. (**A**): Schematic showing preabsorption of sera with one of the conformational probes as shown (blocking), followed by binding to labeled C47/IAPP probe (binding). (**B**): Graphs showing binding of T1D patient sera to labeled C47/IAPP (left) or C58/IAPP (right) probes after preabsorption with increasing concentrations of either unlabeled C47/IAPP (solid lines) or C58/IAPP (dashed lines) probes. Duplicate values were averaged and expressed as CPS on a Base 10 log scale. Affinity absorption was measured three times/sera, and representative data are shown. (**C**): Graph showing binding of T1D patient sera to labeled C47/IAPP after preabsorption with C47/IAPP (solid red circles), cyclic peptide 1 (blue solid squares), cyclic peptide 2 (blue open squares), or nonspecific (BSA) control (open circles). (**D**): Graph showing binding of T1D patient sera to labeled C47/IAPP after preabsorption with C47/IAPP (solid red circles), insulin (green open triangles), proinsulin (gray open hexagons), or nonspecific (PBS) control (black open circles).

**Figure 4 diagnostics-13-02859-f004:**
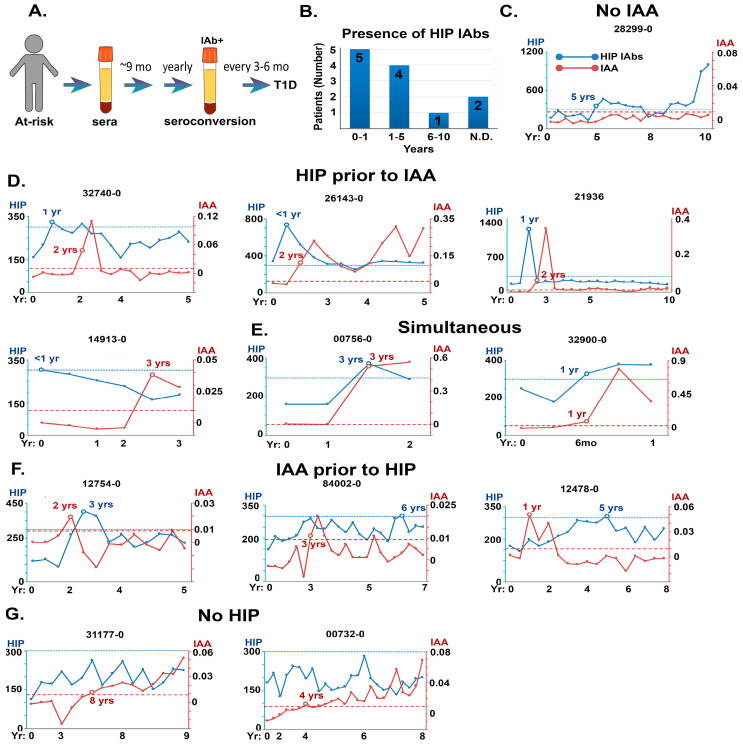
Prevalence and relative time of appearance of HIP IAbs in sera from prospective pre-T1D patients in the DAISY study. (**A**): Schematic showing timeline of serum monitoring for DAISY patients. (**B**): Bar graph showing DAISY subjects binned according to time of HIP IAbs appearance. (**C**–**G**): Timeline comparisons of appearance of HIP IAbs and insulin autoantibodies (IAAs) in selected DAISY patients prior to T1D onset. *y*-axis is CPS for HIP IAbs on the left, IAA on the right. Lower cut-off for HIP IAbs = blue line, IAAs = red line. Time course of the study (in years) is indicated on the *x*-axis. C: HIP IAbs only detected; D: HIP IAbs appear before IAAs; E: HIP IAbs are concurrent with IAAs; F: HIP IAbs appear after IAAs; G: or do not reach significant levels relative to IAAs.

**Table 1 diagnostics-13-02859-t001:** Demographic data of cohorts monitored in this study.

COHORT	*n*	M	Age Range *	F	Age Range *
ASK Controls	96	61	2.2–18	35	2.0–16.8
T1D Linear Probes	95	49	2.3–17.9	46	1.0–17.8
New Onset T1D	269	142	1.3–17.8	127	1.0–17.8
T1D DQ-2/8	89	48	2.9–17.9	41	2.9–17.7
T1D DQ--X/X	84	44	2.9–17.7	40	1.1–17.8
T2D	112	58	9.9–17.9	54	10.7–18
DAISY Subjects	24	11	0.75-T1D	13	0.75-T1D
**BLOCKING**					
**Conformational Probes**			**Cyclic**		
**Subject ID**	**Gender**	**T1D Onset ***	**Subject ID**	**Gender**	**T1D onset ***
571068	F	17.4	562158	F	7.2
572144	F	14.7	563066	F	3.8
577584	F	11.4	564288	F	12.2
578399	M	17.2	564305	F	8.0
581801	M	13.9	564490	M	16.0
137-D2331	F	10.3	593429	F	17.3
CS373	M	1.3	561898	F	14.9
548300	F	15.6	815041	M	pre-diabetic, 1.9
552543	F	12.2	827086	F	pre-diabetic, 7.7
551519	M	14.8	837780	F	pre-diabetic, 12.1
544533	M	12	846582	M	pre-diabetic, 2.2
561898	F	14.8	800578	M	pre-diabetic, 0.8
564490	M	16	804795	M	pre-diabetic, 2.0
564305	F	8	847151	M	pre-diabetic, 16.1
593429	F	17.3	606198	M	9.7
562158	F	7.2	603511	M	11.8
579311	F	12.6	600322	M	11
563066	F	3.8	548300	F	T2D, 15.6
564288	F	12.2	552543	F	T2D, 12.2
605431	M	6.0	137-D2331	F	Healthy control, 10.3
			CS373	M	Healthy control, 1.3
102-D2301	M	12.5			
D827086	F	7.7			
606198	F	9.7			
846926	F	14.3			
844775	F	13.2			
581375	F	15.9			
593237	M	17.9			

Cohort data include population size, gender, and age range of T1D onset. Blocking data: subjects used in blocking studies are listed individually via gender and age of onset of T1D. Other clinical information for these individuals is not available. *n*, number; M, male; F, female. * Indicates years.

## Data Availability

All protocols, reagents, and data are available upon request.

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
