# Peer review of "Identification of Autoantibodies to a Hybrid Insulin Peptide in Type 1 Diabetes"

_diagnostics, 2023, doi:10.3390/diagnostics13172859_

Round 1

Reviewer 1 Report

The Manuscript entitled "Identification of Autoantibodies to a Hybrid Insulin Peptide in Type 1 Diabetes" discusses about the type 1 diabetes Type 1 which is a chronic autoimmune disease targeting the insulin producing beta- cells of the pancreatic islets. In the present study, authors claim the hybrid antigens could be identified as biomarker for autoantibodies in T1D and prediabetic patients. Authors show that sera from T1D patients contain antibodies that bind larger hybrid proinsulin:IAPP probes, but not proinsulin or insulin. There are certain queries related to the autoantibodies. The Manuscript is good, but need improvement to be considered for this journal-

Comments

1. The manuscript should be reviewed for the english language and grammar throughout the manuscript.

2. GAD-65 is one of the marker for the detection of type 1 diabetes, the reference for the same should be cited in the introduction section-

Immuno-chemistry of hydroxyl radical modified GAD-65: A possible role in experimental and human diabetes mellitus. IUBMB Life. 2015 Sep 11;67(10):746-56.

3. Statistical analysis should be described in little more detail.

4. Why and how did authors believe that the sera from T1D patients contain antibodies that bind larger hybrid proinsulin:IAPP probes?

5. Authors should also explain how the new antibodies which are formed in type 1 diabetes are formed? 

English language is to be improved.

Author Response

Dear Reviewer 1,

Thank you for providing us with the opportunity to submit the revised manuscript “Autoantibodies Bind a Hybrid Insulin Peptide in Type 1 Diabetes” (Ref:). The manuscript has been edited to include additional text and information to satisfy the reviewers comments. We appreciate the reviewers’ helpful comments in making this a better manuscript. Please see our responses below.

  1. The manuscript should be reviewed for the english language and grammar throughout the manuscript.

We have reviewed the manuscript and corrected English language and grammar errors. In particular, the script has been run through the DEEPL WRITE IA transductor to make English language, grammar and spelling corrections. Please see corrections highlighted in blue.

  1. GAD-65 is one of the marker for the detection of type 1 diabetes, the reference for the same should be cited in the introduction section-

Immuno-chemistry of hydroxyl radical modified GAD-65: A possible role in experimental and human diabetes mellitus. IUBMB Life. 2015 Sep 11;67(10):746-56.

We have included this citation in the Introduction and in References.

  1. Statistical analysis should be described in little more detail.

A sentence has been added to describe the Welch’s T-test in greater detail. Please see this under Methods/Statistics page 7.

  1. Why and how did authors believe that the sera from T1D patients contain antibodies that bind larger hybrid proinsulin:IAPP probes?

The intermediate molecules for HIP formation remain unknown.  Our published mass spectrometry data was performed using proteolytically cleaved molecules to allow them to “fly” better in mass spec experiments to yield interpretable spectra. The Cpep/IAPP2 HIP was observed in granules, crinosomes and lysosomes by Unanue’s group (now lead by XiaoXiao Wan), but the HIP peptide is relatively short and may represent a fragment of a larger protein. We know from our history of work with autoantibody assay development that most antibody epitopes are conformational. We constructed candidate molecules to reflect what may form in vivo. We have added text in the manuscript (page 9) to explain this rationale.

  1. Authors should also explain how the new antibodies which are formed in type 1 diabetes are formed? 

Several laboratories currently study the mechanism of HIP formation. We have added text and citations (page 9) to include the two major investigations to date. This includes our mass spectrometry collaborator Thomas Delong and other labs.

Reviewer 2 Report

Gentili Autori

in allegato i commenti 

Cordiali saluti

Author Response

Dear Diagnostics Reviewer 2,

Thank you for providing us with the opportunity to submit the revised manuscript “Autoantibodies Bind a Hybrid Insulin Peptide in Type 1 Diabetes” (Ref:). The manuscript has been edited to include additional text and information to satisfy the reviewers comments. We appreciate the reviewers’ helpful comments in making this a better manuscript. Please see our responses below.

Introduction: Authors are invited to integrate the introduction with more details about the role of viral and bacterial infections in diabetes explaining the role of Mycobacteria and Human Endogenous Retrovirus in TD1.

We have included a paragraph with citations in the Introduction to elucidate the impact of viral and bacterial infections to autoimmune diabetes (please see page 3).

Concerning the appearance of islet autoantibodies (IAbs) in the circulation, there are different studies that showed the role of Mycobacterium avium subspecies paratuberculosis (MAP) and HERV-W in T1DM etiopathogenesis in children from Sardinia, supporting for the first time the hypothesis that it may be possible that both pathogens contribute to the onset of T1DM. They found also a correlation between anti- HERV-W and proinsulin (PI) autoantibodies in the sera of children with T1DM collected at different times after onset. Other than that, researchers found for the first time a significant difference in the antibody response against HERV-K between T1DM and HCs and at T1DM onset (PMCID: PMC9607583).

We discuss these findings in the introduction (please see page 3).

Materials and Methods. Please explain which test did you use for determining the power of the research study and the result obtained by the analysis. Report also the demographic and clinical features of all subjects involved in the present study.

To initially determine the potential of any peptide construct as a valid autoantigen, we examined their binding by sera from 96 new onset T1D patients and 96 controls using our ECL assay. Those antigen peptides that exhibited binding toantibodies were considered viable candidates and were then investigated using our complete new onset patient cohort (n=269). This pilot study was designed to address the feasibility of HIP conformational antigens as T1D B cell autoantigens. Future research will focus on extracting genuine in vivo antigens for the development of patient screenings potentially for clinical use.

We have incorporated the islet autoantibody status of the individuals in the new onset T1D cohort and stratified the data with respect to HLA genotype (please see page 10). Beyond gender, age, age of onset no information is available for these individuals. This has been stated (please see Table I).